# R-RDSP: Reliable and Rapidly Deployable Wireless Ad Hoc System for Post-Disaster Management over DDS

**DOI:** 10.3390/s24227259

**Published:** 2024-11-13

**Authors:** Baber Jan, Adnan Munir, Ayaz H. Khan, Ajmal Khan, Basem Al-Madani

**Affiliations:** 1Computer Engineering Department, King Fahd University of Petroleum and Minerals, Dhahran 31261, Saudi Arabia; g202213840@kfupm.edu.sa (B.J.); g202110550@kfupm.edu.sa (A.M.); mbasem@kfupm.edu.sa (B.A.-M.); 2SDAIA-KFUPM Joint Research Center for Artificial Intelligence, King Fahd University of Petroleum and Minerals, Dhahran 31261, Saudi Arabia; 3Communication and Information Research Center, Sultan Qaboos University, Seeb 123, Oman; a.khan1@squ.edu.om

**Keywords:** post-disaster management, wireless ad hoc network, Data Distribution Service (DDS), real-time communication, Raspberry Pi, emergency response, network reliability, RTI DDS

## Abstract

After natural disasters such as earthquakes, floods, or wars occur, cellular communication networks often sustain significant damage or become impaired. In these critical situations, first responders must coordinate with other rescue teams to communicate essential information to central command and survivors. To address this challenge, we have developed a reliable and rapidly deployable wireless ad hoc system for post-disaster management using Data Distribution Service (DDS) middleware, specifically RTI-DDS, named R-RDSP. The R-RDSP further enhances these metrics, achieving a 14.5% improvement in end-to-end delay and a 20.24% improvement in round-trip delay over the RDSP scheme. The R-RDSP system consists of three main modules: client, relay, and server. Each module connects to others via an ad hoc network, ensuring direct device-to-device communication without relying on existing infrastructure. The client module collects and sends the victim’s location and emergency messages. The relay modules forward these messages across the ad hoc networks, ensuring minimal delay and high reliability. Finally, the server module receives the messages, processes them, and coordinates the response. Leveraging RTI-DDS for reliable message distribution, the system demonstrates robust performance even under challenging network conditions.

## 1. Introduction

Natural disasters around the world have wreaked havoc and taken away many valuable people’s lives. After a disaster, it is very important to save the victims and survivors. After earthquakes and flooding, traditional means of communication have had a negative impact or may not be available. Everyone who wants to contact the rescue team as soon as possible and ask for help must have a reliable communication method. It is very important to respond promptly to requests from blocked people to get an overview of general injuries, relocation methods, and emergency needs [1,2,3]. Setting up network components, such as relays and access points, in a way that allows for the creation of temporary networks as needed can help alleviate communication challenges. This involves having a network infrastructure that can be quickly deployed to facilitate emergency operations, including coordinating efforts between helicopters and ground response services, ultimately leading to saving numerous lives.

In the past decade, data distribution systems that ensure Quality of Service (QoS) levels, like the data-centric Data Distribution Service (DDS) standard specification, have become increasingly popular and successful. RTI DDS comes up with multiple QoS checks (e.g., time/content-based filtering, history, deadline, durability) and many more [4]. QoS is dynamic and can be changed at run-time according to the requirement of the application. Pub/sub middleware consists of three main units: Publisher, Subscriber, and Pub/Sub-Service, as in Figure 1. The pub/sub service acts as the middleware hub, responsible for managing publications and subscriptions.

### Contribution

As in our previous work [5], we developed an RDSP (rapidly deployable system for post-disaster) system to minimize the average latency for sending the rescue teams information about the victim’s whereabouts to the control server. The dynamic nature of the RDSP system is a key challenge regarding the scalability and reliability of the system. Our objective is to leverage DDS’s established performance [6] in enabling dependable and effective communication in distributed systems. First and foremost, we aim to curate a system that maximizes the publish/subscribe middleware’s overall performance while simultaneously guaranteeing reliable data stream delivery.

By integrating these two systems, the goal is to enhance the scalability and reliability of the RDSP system by mitigating the complexity associated with managing dynamic data streams. To ensure the R-RDSP architecture can perform effectively under varying workloads, it is essential to develop systems that can dynamically adjust to shifts in data flow patterns. Additionally, leveraging DDS’s advanced capabilities such as its support for Quality of Service (QoS) policies and built-in fault tolerance further strengthens the reliability of the R-RDSP system against potential disruptions. These combined strategies provide a robust foundation for real-time data stream processing, designed to meet and surpass the demands of highly dynamic environments while adhering to stringent reliability and performance requirements.

## 2. Related Work

TinyDDS is a lightweight and partial porting of DDS middleware to touch platforms, especially for platforms with limited access resources. However, TinyDDS, as it stands, does not support data reliability. Al-Roubaiey et al. [7] extended the DDS Data Reliability Service and integrated it with TinyDDS, to create a robust TinyDDS. They also provide a prototype and a full reliability assessment. Functions that are taken into account include the number of transitions, the number of editors, and many other network parameters. They proposed the RTDDS protocol, with its level sensors collecting data every second after or at an appropriate predetermined time. For instance, in fire or toxic gas detection systems, sensors periodically collect data. When a reading exceeds a set threshold, it is marked as “reliable”, indicating an alert or critical event. Otherwise, the reading is classified as “best-effort”, representing normal or non-critical conditions.RTDDS provides three reliable QOs: best-effort QOs, fully reliable QOs, and partially reliable QOs. However, the dynamic network with different topologies is not considered.

In the study [8], Middleware was implemented based on the proposed architecture, serving as a backbone for the network and an autonomous distributed system. The Emergent Distributed Bio-Organization (EDBO) model was introduced and formulated through extensive research into emerging phenomena in artificial distributed systems (ADS). In designing middleware tailored for Cyber-Physical Systems (CPS), the EDBO model allows for multiple suitable options to be incorporated into the final system. An artificially distributed system needs to be designed as a multi-agent, enabling autonomy and fostering innovative ideas, as proposed by the EDBO model. Middleware effectively realizes various scenarios aligned with this intelligence-driven approach. The strength of this system lies in its ability to autonomously adapt, with the EDBO model serving as the foundation for self-organization within the artificial distributed system, facilitating connections and interactions between people and technology.

Alshinina et al. [9] proposed Wireless Sensor Network (SWSNM) middleware based on an automated and generative adversarial network algorithm. This generative algorithm incorporates two networks including the discriminator network (D) and the Generator network (G). This system displays verified analysis of data that is transmitted protectively across the WSN. The system is accomplished with experiments in Python with Keras. The results of SWSNM show it improves the security and accuracy of data from trespassers. In addition, the Algorithm SWSNM has higher output, less overall latency, and appreciably less power consumption versus a similar traditional approach. An intelligent security for GAN-based WSN middleware was implemented to enhance traditional middleware, regarding the security mechanisms dealing with disparate Properties of sensor nodes and only filters and passes real data.

Andreas et al. [10] have proposed an ad hoc surveillance network that works on threshold triggering. The user measures threshold levels to place devices. Once the link quality drops below this threshold, the user must drop a new relay. An ad hoc network is built based on wireless devices powered by a battery. These wireless devices with network connectivity are called drop units. To ensure adequate coverage, excessive use of dropped units seems obvious but this solution leads to undesirable effects, such as interference. To solve this problem, the authors have proposed rules for deploying an optimal number of drop units. However, the amount of time each unit took for message processing is not mentioned.

J. Chen et al. [11] introduced R-COPSS, a revolutionary improvement for content-centric publish/subscribe systems that emphasizes dependability and better flow and congestion control. With the help of R-COPSS, publishers may make use of content-centric networks more effectively by getting rapid feedback from subscribers. Subscribers can also use NDN for local repair to guarantee dependable delivery of content. Their hierarchical technique chooses some subscribers to provide feedback, while others provide periodic summaries. This is in contrast to traditional methods where all subscribers generate feedback per packet. By using this method, the publisher’s effort is not only decreased but it is also no longer required to modify sending rates in response to the slowest subscriber. Based on preliminary results, R-COPSS outperforms existing methods in terms of total throughput and fairness to competing flows. However, no method is defined for ways to lessen the amount of unnecessary traffic on the minority to decrease the overall strain on the network and improve service for the other flows.

## 3. Methodology

The methodology outlines the systematic approach and technical implementations used to establish a robust communication network designed for post-disaster management. This section details the network configuration, DDS architecture, quality of service specifications, and the communication flow within the system, as depicted in Figure 2. Each subsection describes specific components and their interactions to ensure reliable emergency communication.

### 3.1. Network Configuration

To create a robust and efficient communication network for post-disaster management, we implemented an ad-hoc network setup on all devices. Each device was set to operate in ad-hoc mode, enabling direct device-to-device communication without relying on any central infrastructure. This decentralized approach ensures that the network remains functional even if parts of it are damaged or disconnected, which is crucial in disaster scenarios [12].

#### 3.1.1. Ad-Hoc Network Setup

The network configuration involved setting each device’s wireless interface to ad-hoc mode with a common SSID and the same WiFi channel. Each device was configured to generate a unique IP address based on its MAC address, ensuring minimal IP conflicts and simplifying network management.

#### 3.1.2. Continuous Device Discovery and Dynamic Network Management

Upon establishing the ad-hoc network, each device initiates a continuous scanning process for the common SSID and exchanges beacon frames to detect and connect with other devices on the set WiFi channel. This ongoing scanning not only facilitates the dynamic discovery of new nodes but also ensures seamless incorporation of these nodes into the network. As devices broadcast their presence, the network’s routing tables are updated, allowing for immediate adjustments to the network topology. Additionally, devices that cease to transmit beacon frames are promptly removed from routing tables, maintaining the network’s integrity and resilience. It is important to note that no specific routing algorithm or protocol is used; the routing table is solely for managing devices within a single ad-hoc network or WiFi zone. This adaptive mechanism is crucial for maintaining a robust communication framework in the unpredictable and often chaotic conditions typical of post-disaster environments, ensuring that the network can dynamically adjust to changes, with devices joining and leaving as necessary.

### 3.2. DDS Configuration and Quality of Service

The Data Distribution Service (DDS) architecture was configured and set up using Real-Time Innovation (RTI) DDS to facilitate reliable and efficient data exchange, which is crucial for emergency response scenarios.

#### 3.2.1. DDS Architecture

The DDS configuration includes a single domain, where all communication occurs. Within this domain, there are two participants:Relay: Acts as a publisher by sending and forwarding emergency messages, and as a subscriber by receiving messages from other devices.Server: Acts as a publisher by sending acknowledgments or replies and as a subscriber by receiving all incoming messages.

The communication between participants revolves around a single topic. This topic is used for both sending emergency messages from clients to servers (including relay forwarding) and for sending responses from servers to clients (again, including relay forwarding), it can be shown in Table 1. The data type includes several fields:

Message: The content of the emergency or response message.MacAddressesOfVisitedRelays: A list of MAC addresses. The first address is the sender’s address. Each relay adds its address to this list. This helps track the message path and prevent message loops.TTL (Time To Live): Ensures messages do not propagate indefinitely.Latitude and Longitude: Coordinates of the message origin to facilitate precise location tracking.DestinationAddress: Identifiers for the intended recipient of the message.RelaysMessageTimeStamp: Timestamp marking the message’s creation.

#### 3.2.2. Quality of Service (QoS) Settings

The DDS Quality of Service (QoS) settings were carefully selected to ensure robustness and reliability in message delivery while taking into account the limited resources of the devices within the disaster management context. The chosen settings, such as Volatile and KeepLast, might typically be seen as vulnerabilities due to their lack of data persistence and limited message history. However, our network design supports multiple paths from the client to the server, providing redundancy that compensates for these vulnerabilities and allows us to conserve resources on devices with limited capacities. The QoS settings are detailed in Table 2:

#### 3.2.3. DDS Discovery and Network Management

Building upon the ad-hoc network setup, DDS utilizes its built-in discovery mechanisms to identify active participants within the network:Participant Discovery: Utilizes the ad-hoc network for initial detection of relays and servers via multicast on designated ports.Endpoint Discovery: Manages the detection and engagement of data publishers and subscribers, facilitating efficient data routing based on current network topology.

Periodic heartbeats and acknowledgements help maintain an updated view of network participants, enhancing the system’s ability to adapt to dynamic network conditions such as participants joining or leaving the network.

### 3.3. Communication Flow in the DDS Network

Emergency messages in our system flow from clients through relay nodes to a central server. The server processes these messages and coordinates rescue responses. To ensure reliable delivery, each system component implements specific verification steps. We explain this communication process in four phases: message initiation, relay forwarding, server processing, and acknowledgment handling.

#### 3.3.1. Initiation at the Client

In an emergency, the user interacts with the client device, which functions as both a client and a relay node, to send an emergency message. The following steps outline the communication process:User Interaction: The user initiates the client and enters the emergency message.Message Preparation: The client gathers vital information including its own MAC address, geographic coordinates, and a timestamp.Message Construction: The emergency message is assembled with critical data fields:Message: Content of the user’s emergency communication.MacAddressesOfVisitedRelays: Initiated with the client’s MAC address.TTL (Time To Live): Set high to ensure the message’s propagation through the network.Latitude and Longitude: Geolocation of the message origin.DestinationAddress: Set to “SERVER” to direct the message to the server.RelaysMessageTimeStamp: Timestamp marking the message’s creation.Publishing the Message: The message is transmitted over the DDS network using a DataWriter targeted at the *EmergencyMessage* topic.

#### 3.3.2. Forwarding at Relay

As the message propagates through the DDS network, it encounters relay nodes that handle the forwarding process:Message Reception: Relays continuously monitor for incoming messages on the *EmergencyMessage* topic.Message Processing:Checks TTL to ensure the message is still viable.Verifies that its own MAC address is not already listed in *MacAddressesOfVisitedRelays* to prevent looping.Message Forwarding: If valid, the relay appends its MAC address, decrements the TTL, and queues the message for further propagation towards the server.

#### 3.3.3. Response and Acknowledgment at Server

The server, upon receiving a message, processes and responds:Message Reception: Continuous listening on the *EmergencyMessage* topic.Message Verification and Logging: Confirms message integrity (TTL and MAC address validation) and logs the message for records and potential action.Acknowledgment Creation: Crafts an acknowledgment message that reiterates the original message’s data and additional server feedback.Publishing the Acknowledgment: Sends the acknowledgment back through the DDS network, ensuring the client receives confirmation of message delivery and processing.

#### 3.3.4. Acknowledgment Reception at Client

Finally, the client receives the acknowledgment, completing the communication loop:Listening for Acknowledgments: The client device monitors incoming messages on the *EmergencyMessage* topic.Processing Acknowledgments: Upon receiving an acknowledgment, the client checks the original message’s timestamp against outstanding messages.User Notification: Confirms receipt and informs the user of the successful delivery and acknowledgment of their emergency message.

Building on these communication phases, each component implements mechanisms to handle messages in the multi-path network. When a client creates an emergency message, it generates a timestamp that serves as a unique identifier for this message. This timestamp is included as the RelaysMessageTimeStamp field in the message before transmission. This same timestamp must be included by the server in its acknowledgment/reply message in the same field to identify which emergency message it is acknowledging. The client tracks each emergency message it sends by maintaining a list of these timestamps. When an acknowledgment message arrives, the client checks its RelaysMessageTimeStamp field against its list of sent message timestamps. Once the client receives an acknowledgment with a matching timestamp, it marks that message as confirmed and discards any future acknowledgments containing the same timestamp. Hence, only accepting the first instance.

Each relay node performs two specific validations on every received message. First, it verifies that the TTL field has not reached zero. Second, it checks that its own MAC address is not present in the message’s MacAddressesOfVisitedRelays field. Through these two validations, when a relay receives multiple copies of the same instance of a message from different neighboring relays or clients, only the first received copy is processed and forwarded.

At the server, each source from which it receives a message is assigned a unique ID, and the server keeps a list of message IDs (RelayMessageTimeStamps) it has acknowledged for each source/client. When the server receives an emergency message, it checks if that message’s ID exists in the client’s acknowledged list. For a new ID, the server processes the message and sends an acknowledgment. For timestamps already in the list, the server drops the message, as it has already been processed from this sender.

## 4. Experimental Setup and Evaluation

This section delves into the experimental setup of the Revised Real-time Disaster Support Protocol (R-RDSP), detailing the deployment scenario and evaluating its performance against our earlier RDSP system [5]. We focus on the practical application and effectiveness of R-RDSP in simulated disaster response scenarios, assessing both technical configurations and operational capabilities.

### 4.1. Experimental Setup

The detailed configurations of the network and DDS settings, along with device specifics, are provided in Table 3:

### 4.2. Deployment Scenario

The R-RDSP was deployed in a controlled environment to simulate a real-world disaster scenario and test its effectiveness under operational conditions. The deployment was carried out on the football ground of King Fahd University of Petroleum and Minerals (KFUPM), as shown in Figure 3. The football ground, spanning approximately 10,330 square meters, was chosen for its open space and minimal interference, which are ideal for testing wireless communication systems like the R-RDSP. Multiple Raspberry Pi devices were set up as relay nodes in various configurations to create different paths for data transmission from the client to the server, simulating potential real-world communication routes that could be established post-disaster. Relay nodes in the RDSP system are dynamically deployed by the rescue team members during emergencies. The rescue team moves forward and places relay devices at equal intervals of about 90 m, using GPS modules to guarantee spacing. Relay devices allow communication between the client devices of rescue teams and the server device placed at the scene of the incident.

To ensure comprehensive coverage and test the system’s robustness, several relay nodes were placed strategically across the field. The configuration included different paths such as S-X-Y-C, S-X-Z-N-C, among others, where ’S’ stands for the server, ’C’ for the client, and ’X’, ’Y’, ’Z’, ’N’ represent various relay nodes. This setup allowed the evaluation of the system’s functionality across multiple pathways, ensuring that data could be reliably transmitted across both the shortest and alternative routes, depending on the network conditions. The parameters used in conducting the experiments are summarized in Table 4:

### 4.3. Evaluation of R-RDSP

While these network-level metrics are valuable, this work focuses on using DDS middleware as an abstraction layer to handle network complexities. We chose DDS specifically because it provides reliable message delivery and network management through its Quality-of-Service settings, allowing us to concentrate on building efficient emergency communication systems. We implemented a broadcast-based approach to ensure message delivery through multiple paths, crucial for disaster scenarios where network links can fail unpredictably. In our experimental setup with five paths and seven relays, we observed approximately 3–4 message copies per transmission due to broadcast. However, rather than optimizing at the packet level, we demonstrate in Section 4.3.3 how our system manages this overhead at the application level through timestamp filtering and MAC address tracking. This abstraction lets DDS handle the network efficiency while we ensure our system meets critical emergency response needs.

To evaluate and compare the performance of the Reliable-Rapidly Deployable Wireless Ad Hoc System for Post-Disaster Management (R-RDSP), with simple Rapidly Deployable Wireless Ad Hoc System for Post-Disaster Management (RDSP), several key performance metrics are used. These metrics included the distance covered by relay nodes, end-to-end delay, round-trip delay, and network overhead. Each metric provides insights into the system’s efficiency, reliability, and practical applicability in a real-world disaster scenario.

#### 4.3.1. Distance Covered

The distance covered by relay nodes is a crucial metric in disaster scenarios where extensive coverage is necessary. Figure 4 illustrates the distances covered by various relay paths set up in the deployment scenario. According to the scenario, Path 1 (C-Y-X-S) achieved a total of 270 m using two relay nodes, while Path 2 (C-N-Z-X-S) extended to 360 m with three relay nodes. Other paths like Path 3 (C-N-Z-S), Path 4 (C-N-Z-P-S), and Path 5 (C-R-Q-P-S) covered distances of 360 m, 250 m, and 360 m, respectively.

#### 4.3.2. End-to-End and Round-Trip Delays

End-to-end delay measures the time taken for a message to travel from the client to the server, while round-trip delay measures the time for a message to travel to the server and back to the client. These delays are influenced by the number of relays and the total distance covered. Our experiments show that R-RDSP improves upon the previous RDSP [5] system by reducing the end-to-end delay by 14.5% and the round-trip delay by 20.24%, as shown in Figure 5 and Figure 6.

#### 4.3.3. Network Dynamics, Path Redundancy, and Overhead

As described in Section 3.2.1, R-RDSP utilizes a single topic where each device publishes and subscribes to all message types (emergency messages and acknowledgments). This design choice leverages DDS’s broadcast/multicast mechanism where each published message reaches all devices within WiFi range. In disaster scenarios where network infrastructure is often damaged or unreliable, this provides critical features: message delivery is guaranteed through multiple paths ensuring communication even when network links fail unpredictably, messages traveling through all available paths simultaneously ensure delivery through the fastest route for time-critical responses, and devices can dynamically join or leave the network without reconfiguration, enabling rescue teams to rapidly deploy or relocate devices as disaster conditions change.

Broadcasting creates network overhead through duplicate messages and potential message loops. As detailed in Section 3.3, each network component uses timestamp identification, TTL checks, and MAC address verification to handle these issues. Clients track message IDs (timestamps) and process only the first acknowledgment they receive for each sent message. Relays check TTL and MAC address lists to prevent loops, forwarding only the first instance of each message they receive. The server maintains a message history per source and processes only new messages, dropping any duplicates. Through these mechanisms at each point, our system prevents unnecessary message propagation and loops while automatically selecting faster paths. This efficient handling of broadcast overhead enables our system to maintain the reliability and speed crucial for disaster response communication.

## 5. Conclusions

Wireless Ad-hoc Networks (WANETs) play a crucial role in facilitating data transfer across various applications. Middleware serves as a mediator between applications and WANETs, addressing concerns related to energy consumption, communication, and security. This article aims to establish reliable communication in multi-hop wireless ad-hoc networks through middleware for disaster management systems. In this study, we reduced the average waiting time for request messages that are sent and communicated between victims and servers. Additionally, this work outperforms our previous RDSP [5] work, especially for the network overhead and dynamic joining and leaving time by using DDS. The end-to-end delay and round trip delay are improved by a factor of 14.5% and 20.24%, respectively. In future work, we will reduce the discovery time for nodes. We can also improve some of the DDS quality of services for better and more reliable communication. The realtime data including audio and files will be considered for communication between victims and servers.

## Figures and Tables

**Figure 1 sensors-24-07259-f001:**
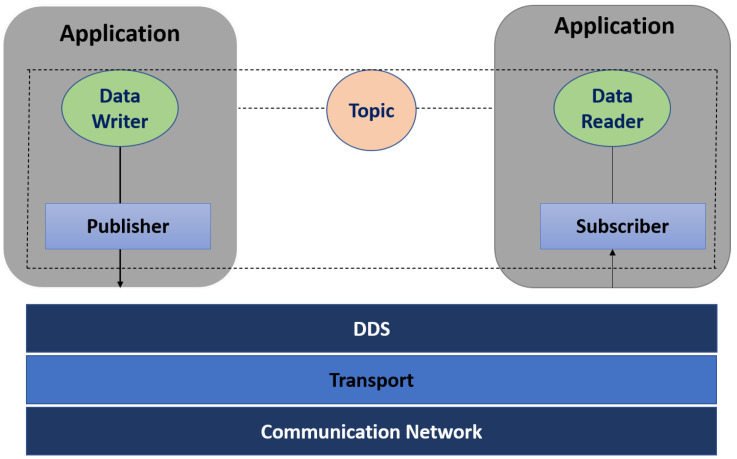
DDS Architecture.

**Figure 2 sensors-24-07259-f002:**
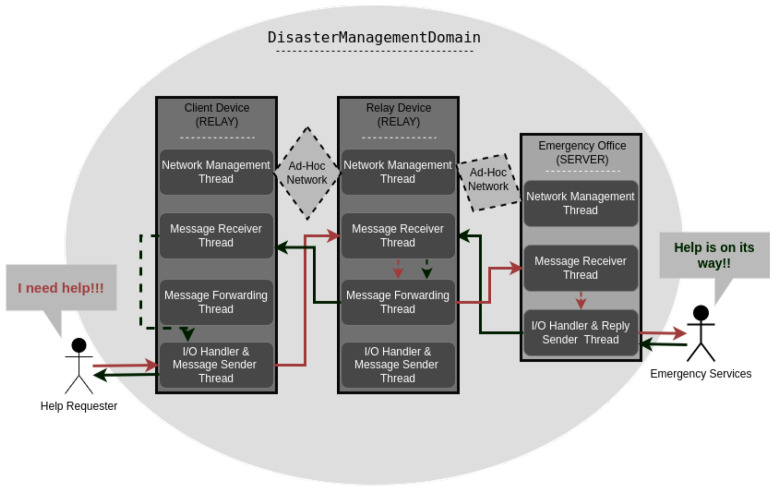
Architectural Flowchart of the Disaster Management System.

**Figure 3 sensors-24-07259-f003:**
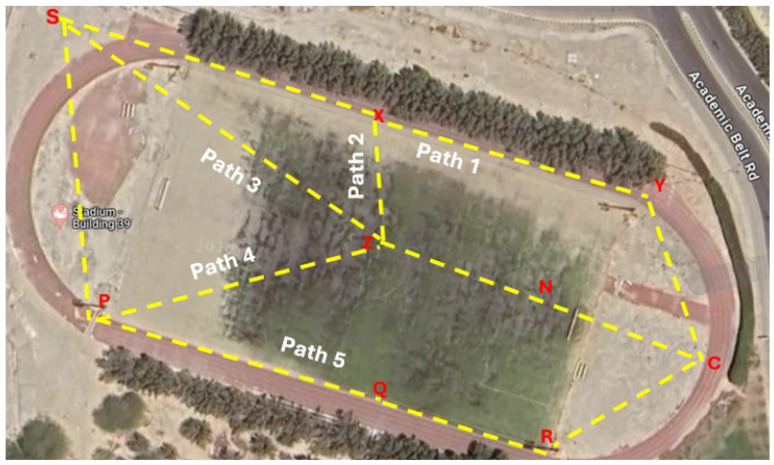
Deployment Scenario of R-RDSP. The paths represented in the scenario are: Path 1 (C - Y - X - S), Path 2 (C - N - Z - X - S), Path 3 (C - N - Z - S), Path 4 (C - N - Z - P - S), and Path 5
(C - R - Q - P - S).

**Figure 4 sensors-24-07259-f004:**
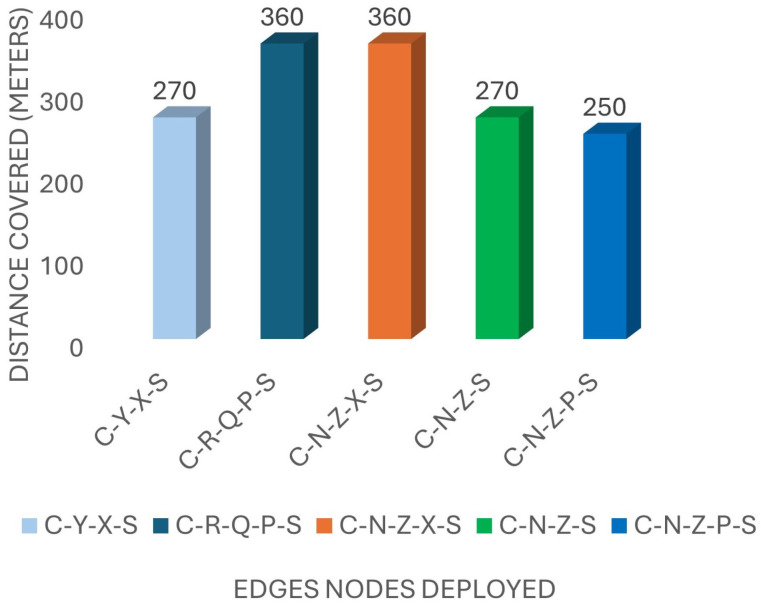
The total distance covered by relay devices.

**Figure 5 sensors-24-07259-f005:**
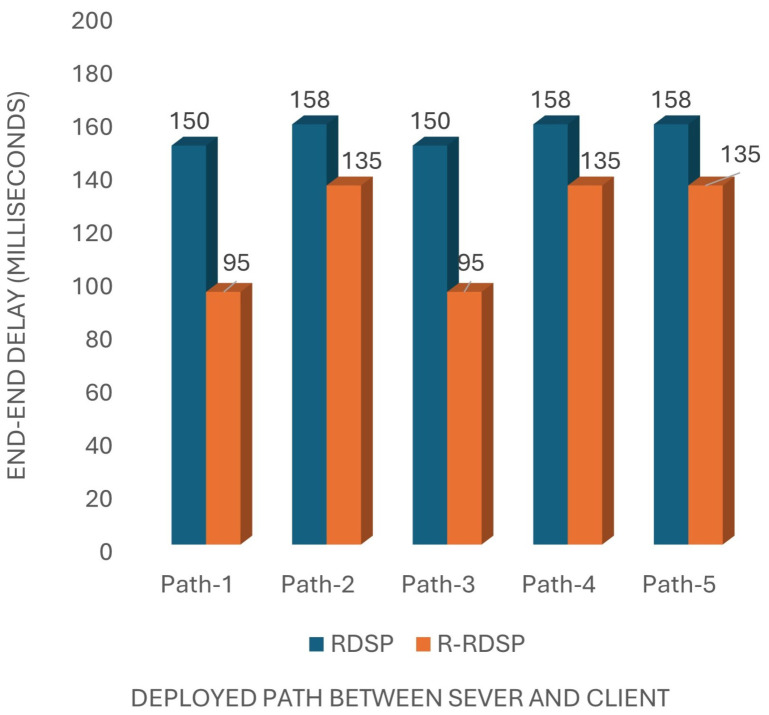
End-to-end delay comparison between R-RDSP and RDSP.

**Figure 6 sensors-24-07259-f006:**
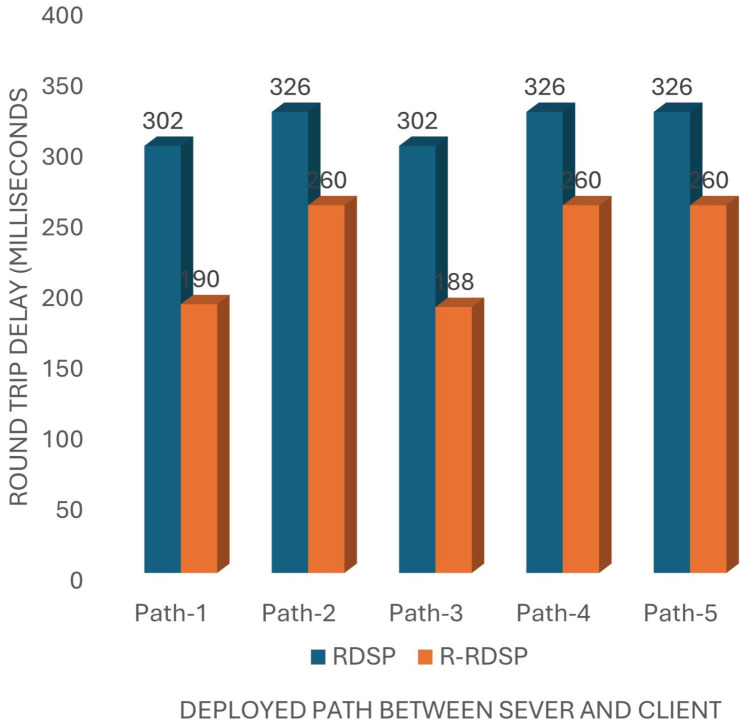
Round-trip delay comparison between R-RDSP and RDSP.

**Table 1 sensors-24-07259-t001:** DDS Communication Schema: Emergency Message Routing and Response Handling.

Participant	Role	Topic	Data Type Fields	Message Flow
Client	Publisher	Emergency	Message MacAddressesOfVisitedRelays TTL Latitude Longitude DestinationAddress RelaysMessageTimeStamp	Client -> Relay -> Server
Relay	Publisher, Subscriber	Emergency, Response	Message MacAddressesOfVisitedRelays TTL Latitude Longitude DestinationAddress RelaysMessageTimeStamp	Client -> Relay -> Server, Server -> Relay -> Client
Server	Publisher, Subscriber	Emergency, Response	Message MacAddressesOfVisitedRelays TTL Latitude Longitude DestinationAddress RelaysMessageTimeStamp	Server -> Relay -> Client

**Table 2 sensors-24-07259-t002:** DDS Quality of Service (QoS) Settings and Their Impact.

QoS Policy	Setting	Impact on Project
Reliability	RELIABLE_RELIABILITY_QOS	Guarantees that messages are delivered without loss, essential for reliable emergency communication.
Durability	VOLATILE	Ensures no outdated messages are stored, only current data is relevant, reducing memory use.
History	KEEP_LAST	Maintains only the most recent message, crucial for timely decision-making in emergencies.
Liveliness	AUTOMATIC	Confirms the presence of publishers, ensuring data is always current and valid.

**Table 3 sensors-24-07259-t003:** Network Settings and Device Configurations.

Configuration	Details
Device Used	Raspberry Pi 4 Model B
Operating System	Raspbian Buster
Programming Language	Python 3.7
DDS Implementation	RTI Connext DDS Connector
SSID	DisManagementNet
WiFi Channel	1
DDS-Domain Name	DisasterManagementDomain
DDS-Topic Name	EmergencyMessage

**Table 4 sensors-24-07259-t004:** The Deployment Parameters and Their Descriptions.

Parameter	Value	Description
Deployment Scenario	KFUPM Football Ground	
Covered Area	10,330 m^2^	The total area covered by the network
Number of Relays	7	Total relays used in the deployment to create redundant paths and ensure network resilience
Number of Paths	5	Different network paths tested to evaluate the network’s ability to find and maintain the most efficient route
Transmission Range	90 m	The maximum distance over which the nodes could communicate effectively
Discovery Interval	5 s	The frequency at which nodes broadcast their presence to maintain updated network topology
Experiment Duration	3 h	Total time for which the deployment was monitored to assess performance and stability

## Data Availability

Data are contained within the article.

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
