# Peer review of "R-RDSP: Reliable and Rapidly Deployable Wireless Ad Hoc System for Post-Disaster Management over DDS"

_sensors, 2024, doi:10.3390/s24227259_

Round 1

Reviewer 1 Report

Comments and Suggestions for Authors

The research presented in the article focuses on improving communication reliability and efficiency in disaster management systems, particularly in rapidly deployable systems (RDSP). The authors aim to enhance the scalability and reliability of their system by integrating Data Distribution Service (DDS) middleware, which is known for its performance in dispersed systems. Their approach seeks to reduce latency in sending critical information about victims to control servers, and they also focus on ensuring reliable data stream delivery in dynamic environments. By utilizing DDS's Quality of Service (QoS) features and fault tolerance, the system is better equipped to handle disruptions and shifting data flow patterns. The results show significant improvements in network overhead, dynamic joining and leaving time, and overall delay reduction, outperforming the authors' previous RDSP implementations.

The conclusions are consistent with the evidence and arguments presented in the paper.

The presented experimentation results demonstrate the effectiveness of the new R-RDSP system and indicate substantial improvements over the previous RDSP system. The figures presented in the article are legible and made according to the best presentation practices.

I consider the lack of experiments taking into account the transmission of packets from more than one client to be a certain shortcoming. Furthermore, the scale of the experiments conducted was presented very sparsely. There is information about the experiment duration, but nothing was written about the statistics of the number of packets sent during this period. From what sample were the presented improvement factor percentages of end-to-end delay and round-trip delay calculated? This lack of information makes it impossible to determine whether the results presented are statistically reliable.

Although the article's references are a bit scarce (compared to the references included in the authors' previous article [5]), they strongly refer to the subject of the research undertaken in the article.

Revision suggestions:

- addition of the information about the statistics on the number of packets sent during the experiment.

- lines 194-196 in subsection Communication Flow in the DDS Network  ---> The first statement in the sentence “The communication flow within our DDS-configured disaster management system ensures rapid, reliable message exchange from initiation to acknowledgment, facilitating effective emergency responses” does not explicitly explain the communication flow. In my opinion, such a statement should be placed in the Conclusions rather than at the beginning of this subsection.

- line 200: “The client prompts the user to input an emergency message.” ---> I don't quite understand the intentions behind this sentence. In case of an emergency, does the client decide to kindly prompt a user to ask him to send an emergency message? In case of a disaster, it is a user who experiences the need to send an emergency message, and he does it within his capabilities, arising from the injuries he suffered. The client should only assist if it somehow detects an accident and the unconsciousness of a user, sending the message for him, not to prompt the user to do so. Please rephrase the sentence or make what the authors meant to achieve by prompting user interaction more understandable.

- lines 205 and 209: If the MacAddressesOfVisitedRelays data field is initiated with the client’s MAC address, why does the SourceAddress field duplicate this information?

- lines 249 and 252: The same section name is repeated. Section number 4 contains one initial sentence, which is a paraphrase of the first sentence in section 5. It is necessary to standardize and remove the duplicate text.

Comments on the Quality of English Language

Minor errors found in the text:

- line 22: “Everyone who wants to contact the rescue team as soon as possible and ask for help must need some reliable communication way.”---> Everyone who wants to contact the rescue team as soon as possible and ask for help must have a reliable communication method.

- line 64: “They proposed the RTDDS protocol with its level. sensors collect data every second, after or at an appropriate predetermined time.” ---> They proposed the RTDDS protocol, with its level sensors collecting data every second after or at an appropriate predetermined time.

- line 66: “Data can be divided into reliable and more efficient data based on a reading of the sensor for which the sensor is performing network processing to check the reading.” ---> How can data be reliable or efficient? Do the authors have in mind a data transfer that can be reliable/efficient?

= line 68: “When the measured value is greater than the predetermined value (Threshold) , then marks the message as approved.” ---> When the measured value is greater than the predetermined value (Threshold), the message is marked as approved.

- line 69: “RTDDS provides three reliable QOs such as Best best-effort QOs, fully reliable QOs, and partially reliable QOs.” -> RTDDS provides three reliable QOs: best-effort QOs, fully reliable QOs, and partially reliable QOs.

- line 108: “Our hierarchical technique chooses some subscribers to provide feedback, while others provide periodic summaries.” ---> When quoting the technique presented in the article [11], it should be presented as the fruit of the work of the authors of the article [11], which is contradicted by the word Our used in the quoted sentence.

- line 335: “leaving time by suing DDS” ---> using DDS

- line 336: “The end-to-end delay and round trip delay are improved by the factor of 14.5% and 20.24% respectively.”  ---> by a factor of

There are problems with punctuation in the article - some long sentences have too few or no commas. I list a few examples:

- sentence from line 25: “Setting up network components such as relays and access points in a way that allows for the creation of temporary networks as needed can help alleviate communication challenges.”---> Setting up network components, such as relays and access points, in a way that allows for the creation of temporary networks as needed can help alleviate communication challenges.

- sentence from line 336: “The end-to-end delay and round trip delay are improved by the factor of 14.5% and 20.24% respectively.”  ---> The end-to-end delay and round trip delay are improved by a factor of 14.5% and 20.24%, respectively.

Reviewer 2 Report

Comments and Suggestions for Authors

The paper presents a reliable and rapidly deployable wireless ad hoc system for a scenario without network infrastructures, like after the disaster. The proposed scheme R-RDSP outperforms the existing RDSP by significantly reducing delays. The topic and the scheme are valuable considering the high frequency of natural disasters all over the world. However, some necessary discussions are missing from the paper.

The deployment of relay nodes is not discussed. Should the relays be existing nodes? If not, how do we deploy the relays and how do we decide the locations? Also, are the relays dynamic or static?

Did not discuss the routing on the relays, or the path selection approach. If path selection is necessary, i.e., broadcast is used, how to reduce the overhead (replicated data) in the network?

It is necessary to explain the major contribution R-RSDP in the methodology section and how it advances (differs) from the RSDP approach.

Section 4 is unnecessary since it duplicates with Section 5.

Some discussion is required regarding “by processing only the first instance of a message from a given source address, R-RDSP optimizes the path selection dynamically, choosing the fastest route available for message delivery.” For example, how the optimization of path selection spreads to all other locations in the network.

Reviewer 3 Report

Comments and Suggestions for Authors

In the manuscript, the authors have proposed a reliable and rapidly deployable wireless ad hoc system for post-disaster management over DDS. Practical experiments have been conducted in a football court to evaluate the performance of the proposed system. However, some concerns still need to be addressed.

1.     The football ground is an ideal open area. However, the post-disaster area can be harsh, with multiple obstacles and complex communication circumstances. The experimental scenario is not convincing enough.

2.     The post-disaster area is probably hard to access. How do we deploy the relay nodes?

3.     The authors did not mention the power supply of the relay nodes. The post-disaster area can hardly support a wired power supply. How do we power the relay nodes? Wi-Fi is not a protocol designed for low-cost and low-power application scenarios. How do we maintain the lifecycles of the relay nodes?

4.     Could the authors please improve Figure 3 with more labels of the different paths? The paths can hardly tell from Figure 3.

5.     Subsection 5.3.2 provides the comparison of end-to-end and round-trip delays between the current work and their previous work. However, the authors only described the comparison results and did not provide insightful analysis. Besides, the comparison was not conducted with any other works in the literature.

6.     In Subsection 5.3.3, the discussions on the network dynamics, path redundancy, and overhead were conducted verbally without any experimental results.

7.     Section 4 Experimental Setup and Evaluation has only one paragraph.

8.     Section 5 is also entitled Experimental Setup and Evaluation.

Comments on the Quality of English Language

The writing needs to be polished.

Round 2

Reviewer 1 Report

Comments and Suggestions for Authors

All of my proposed corrections have been incorporated into the manuscript. I have no further comments to make.

Reviewer 2 Report

Comments and Suggestions for Authors

Thanks for the response. There are no more comments.

Reviewer 3 Report

Comments and Suggestions for Authors

I have no further comments.

Comments on the Quality of English Language

I have no further comments.